# Social network and tourism consumption by households: Evidence from China

**Genjin Sun**, **Qi Qian, Yanxiu Liu, Bo Pu, Dan Wang** *

School of Business and Tourism, Sichuan Agricultural University, Dujiangyan, Sichuan Province, China

* 1427267736@qq.com

**Data Availability Statement:** All relevant data are within the manuscript and its Supporting Information files.

**Funding:** The authors received no specific funding for this work.

## Abstract

Tourism consumption is not only an important means by which to improve residents' sense of happiness but is also the main way to promote national economic development. In a traditional relational society such as China, it remains unclear how social network affects tourism consumption by households. Here, we evaluated the impact of the social network on tourism consumption by Chinese households using the data of 3254 samples from the China Family Panel Studies. The empirical results from the ordinary least square method showed that the social network promotes tourism consumption, which can be projected to increase by about 28% for every 1% increase in social network strength. This was further confirmed using the instrumental variable method to address the issue of endogenous social network formation, as well as other robustness checks. The impact of the social network on tourism consumption was heterogeneous. Compared with other residents, there were higher positive effects for high-income families, households with a head aged 35–44 years, urban families, and households in eastern China. The quantile regression results revealed that the impact of the social network was weakened with increasing tourism consumption by households. These results are crucial for policymakers, in that they could form good habits of tourism consumption and strengthen tourism market management, especially for the management of tourism negative events in the context of new media.

## 1. Introduction

The level of tourism consumption by Chinese households has continuously increased since the Reform and Opening-up eras [1]. Tourism consumption has played an important role in improving residents' quality of life and promoting national economic development [2–4]. In 2019, Chinese tourism expenditure was 5725.09 billion Chinese Yuan and the per capita tourism expenditure was 944.7 Chinese Yuan, accounting for 16.17% and 4.38% of total consumption and per capita residents' consumption respectively. Thus, these expenditures have increased by 67.42% and 10.23% respectively, since the end of the Twelfth Five-Year Plan in China. With the advent of the mass tourism age, tourism has become an important source of residents' sense of happiness, and tourism consumption has become a stable demand of daily life [5]. Meanwhile, as an important aspect of consumption upgrade, tourism consumption will continue to promote China's economic growth [6].

**Competing interests:** The authors have declared that no competing interests exist.

The factor influencing tourism consumption is an important research topic. Extant studies have classified factors that influence family tourism consumption into four categories, as follows: economic factors [7–10], policy factors [11–13], macroenvironmental factors [14–18], and tourists' own factors [19–25]. Among tourists' own factors, an individual's social network is one of the most important and often neglected factors that influence tourism consumption. A person's social network refers to the stable social relationships with other people from the perspective of social economics, and is formed during people's work and daily lives [26]. Social networks comprise multidimensional and complex interpersonal relationships, such as blood relationships, geographical relationships, business relationships, and interest relationships [27]. It is known that family belief in consumption has a positive effect on the relationship between family travel intention and actual tourism consumption, such as accepting travel recommendations from relatives and friends, and sharing tourism decision information with friends and relatives [28]. In addition, occupation status, marital status of the household head, and community type also influence household tourism consumption [29,30]. In particular, occupation status is a significant socioeconomic discriminating factor in tourism consumption, and families in which the householder has a prestigious job are more likely to participate in tourism and travel abroad than domestically [31].

There is a traditional relational society in China [32], and a person's social network has a strong economic influence [33], greatly affecting household consumption [34,35]. As an important part of family consumption, tourism consumption is the main force to upgrade their consumption for Chinese residents. Tourism consumption is not only becoming an important means of residents' sense of happiness, but also a new driving force for the development of China's national economy. However, few studies have investigated the effect of social network on households' tourism consumption in China. There are several key important questions do not still be answered. Whether social network plays an active role in family tourism consumption? And how does social network influence family tourism consumption? Is there heterogeneous? Therefore, it is necessary to study this issue. And it is the motivation for this study, seeking answers to these questions.

This study explored the effect of social networks on tourism consumption by Chinese households. Data from the China Family Panel Studies (CFPS) were included in the study. The ordinary least square (OLS) and instrumental variable (IV) methods were applied to address the issue of endogenous social network formation, and a variety of heterogeneity analyses were performed. The empirical results obtained using the OLS method revealed that social networks promote tourism consumption by Chinese households, which was further confirmed by the IV method, which also addressed the issue of endogenous social network formation. Subsequent analyses showed that the effect was heterogeneous. Tourism consumption by households with a higher income, a head of the family aged 35–44 years, lived in the city, or located in the eastern regions was more positively affected by social networks than the other studied groups. The results of quantile regression model (QRM) revealed that this influence weakened with increasing tourism consumption. The contribution of the present study is three-fold. First, this study has strong theoretical significance as it investigated the factors influencing tourism consumption by Chinese households, thus expanding the research on tourism consumption from a socio-economic perspective, and providing empirical evidence that allows us to better understand how the market drives increasing tourism consumption demand. Second, the study's conclusions are robust because it adopted representative and time-sensitive microdata of Chinese households, used the IV method to test for robustness, and attempted to overcome the endogenous problem of social network variable selection. Finally, the conclusions from the heterogeneity analysis are of high policy value, are more detailed and targeted, and provide a theoretical basis for promoting tourism consumption in different market sectors.

The rest of the article is structured as follows. The second section states the hypotheses tested in the study. The third section presents the variables, statistical properties, and estimation methods used to analyze the data. The fourth section analyzes the empirical findings, and the fifth section concludes the study with several policy implications.

## 2. Research hypothesis

According to the theory of consumption economics, a person's social network influences tourism consumption via four mechanisms. The first mechanism is the demonstration effect. The spread of tourism willingness through social networks encourages families to imitate each other's behavior [36,37], which results in the "herd effect" and increase tourism expenditure [38,39]. Second, people can pursue achieve "status seeking" behavior or "emotional identity" via tourism activities to maintain their social network [40–43]. This can be called the inductive effect, which is more evident when tourism consumption is highly recognized in a social network. Third, the income effect plays an important role whereby the social network directly promoting family social-networking consumption, and weakening the inhibitory effect of preventive savings motivation and liquidity constraints on household consumption [44]. The social network improves family income via informal finance, employment, entrepreneurship, and so on [45], which enhances the spending capacity for tourism consumption. The final mechanism is the guarantee effect. The social network can act as an informal type of insurance and helps people resist negative influences [46]. People will then have more confidence and competence to pursue greater tourism consumption. The demonstration and inductive effects can stimulate tourism consumption desire, whereas the income and guarantee effects can improve the ability to pay for tourism. In consideration of these effects, households' tourism consumption desire and tourism payment ability are unified, which leads to tourism consumption behavior. Therefore, we made the following hypothesis:

H1: The social network can promote tourism consumption by Chinese households.

Households with different income levels have heterogeneity in their consumption behavior [47,48]. Compared with low-income groups, high-income households have stronger consumption habits, higher marginal propensity to consume, and more active status consumption [49]. Therefore, social networks have different influences on the tourism consumption of households with different income levels. On the one hand, although the importance of family income level cannot be ignored [50,51], families with higher incomes have a smaller elasticity of tourism consumption demand. Moreover, the tourism consumption expenditure by families with higher incomes is more easily affected by the demonstration effect of the social network. On the other hand, high-income families engage in emotional communication and maintain their social networks through leisure vacations and tourist shopping. According to Maslow's need hierarchy theory, tourism consumption demand is a higher-level demand, and also meets physiological and safety needs [52]. For households with a lower income level, income and guarantee effects created by the social network initially influence the low-level demand. In contrast, the income and guarantee effects of the social network usually occur in the high-level demand of tourism consumption for the higher income groups. Based on the above analysis, we put forward the following hypothesis:

H2: Compared with low-income households, the social network has a greater influence on tourism consumption of high-income households.

The factors influencing family tourism consumption decisions are different, at different stages of the family life cycle [53,54]. When children are adolescent, due to economical insolvency, the social network affects daily consumption aimed at young people, but it has little impact on tourism consumption. In the childhood stage, there is a large amount of

consumption expenditure dedicated to raising young children and building families. Although the marginal consumption tendency of families is high in this stage, the tourism consumption effect created by the social network is not apparent. The consumption ability of middle-aged families is significantly increased; as children grow older, the position and proportion of tourism consumption on household consumption increase unprecedentedly. At the same time, the career and social circles of parents are formed. With the stimulation of career development planning, there is an expanding momentum of expenditure. The demonstration effect, income effect, and induced effect caused by the social network matter greatly for all household consumption. As the family enters middle or old age, the family business and social circle have formed, and the income level has stabilized [55]. Therefore, the effect of the social network on family tourism consumption starts to decline. Based on this analysis, we put forward the following hypothesis:

H3: The influence of the social network on tourism consumption changes according to the age groups of Chinese households, and there is an inverted U-shaped relationship between the degree of influence and age.

The different economic development levels and living patterns between urban and rural regions mean that there are differences in social network influences on the tourism consumption of urban and rural households. The long-term urban–rural dual structure in China means that urban households have a higher tourism consumption capacity and stronger tourism consumption habits than rural households [56,57]. Thus, they spend more on traveling, and their elasticity of tourism consumption demand is lower [58]. The social network also has an income effect and guarantee effect on rural households [59]. However, these effects mainly occur in the non-tourism consumption field given the living standards and consumption habits of rural households [60]. The promotion effect of tourism consumption by rural households formed by the social network is weaker than that of urban households. Based on the above analysis, we made the following hypothesis:

H4: The social network has a greater impact on the tourism consumption of Chinese urban households than rural households.

The relational society is a notable feature of China [61]. A vast territory, numerous nationalities, and various local customs have led to regional differences in social network intensity and effect types [62]. Differences in regional economic and social development between the east and west regions of China have, to a certain extent, caused heterogeneity in the consumption habits of Chinese households [63]. The social network influences the formation of households' consumption concept. Groups who live within their means and those who spend beyond their means have different attitudes toward tourism. The differences in economic development and living standards in different geographical regions have led to a regional imbalance in tourism consumption habits and capacity, and the marginal propensity in tourism consumption is also different [64]. Accordingly, the tourism consumption effect driven by social networks also may show regional differences. Thus, we made the following hypothesis:

H5: The impact of the social network on tourism consumption by Chinese households has regional heterogeneity.

According to Keynesian's consumption theory, household income significantly affects household consumption [61]. As an important part of household consumption, the amount of tourism consumption expenditure depends heavily on the household income level in the short term [24]. There is a positive correlation between household tourism consumption expenditure and household income. When the household income level is low, the demonstration effect, income effect, and guarantee effect created by the social network promote households' tourism consumption expenditure. However, a continuous increase in households' tourism consumption expenditure implies an increased family income [65]. The income effect of the

social network is weakened when the marginal propensity to consume is diminished, which results in a continuous decrease in tourism consumption expenditure. Families with higher tourism consumption expenditure have higher economic strength and relatively mature household consumption habits. Tourism consumption occupies an important position in household consumption expenditure, and the economic effect of the social network can be expected to weaken in terms of stimulating the promotion of household tourism consumption. Thus, we made the following hypothesis:

H6: The influence of the social network on tourism consumption by Chinese households gradually decreases with the continuous improvement of family tourism consumption.

## 3. Materials and methods

### 3.1 Data

The primary data were derived from the CFPS, produced by the Institute of Social Science Survey, which were compiled by Peking University. Conducted in 2018, the study included data from 25 provinces, municipalities, and autonomous regions in China, and the survey objects included: individual, family, and community levels (https://opendata.pku.edu.cn/dataverse/CFPS). Based on the requirements of our research, the family questionnaire data and individual data were included. In the CFPS (2018), the sample size at the family level was 14,241 households. The sample without tourism consumption was eliminated from the dataset given that we investigated the influence of the social network on household tourism consumption. Therefore, data from 3,254 households were included in our study.

### 3.2 Ethics statement

The data used in this study were sourced from the CFPS. CFPS is a nationwide, comprehensive social tracking survey project that was designed to provide a data foundation for academic and policy research. To ensure that the rights and interests of respondents participating in the project were protected to the greatest extent, the project team regularly submits applications for ethics review or continuous review to the Biomedical Ethics Committee of Peking University, and conducts corresponding data collection when the ethics review is approved. The ethics review batch number of the project of the CFPS is unified as: IRB00001052-14010, which will not change as a result of different investigation rounds. We obtained the informed written consent.

### 3.3 Variables

**3.3.1 Dependent variable.**   The dependent variable was household tourism expenditure, which was taken as the tourist consumption level of Chinese households. The tourism expenditure of households comes under "tourism expenditure" which refers to the tourism expenditure by households over the past 12 months, including transportation expenses, scenic spot tickets, accommodation expenses, and so on.

**3.3.2 Independent variable.**   The social network was the independent variable, and was measured by gifts of money. Following a study by Yi et al. [44], we used the logarithm of the amount of gift or money given by households and took "gift of money" as a surrogate variable for households' social network. The interaction between households' families, relatives, and friends or others is directly reflected in the attainment and expenditure of gift money. Compared with the attainment of gift money, gift money expenditure is a relatively stable cash flow. Gift money expenditure given by families to relatives and friends is the investment and maintenance of the family social network. To solve the endogeneity problem, we chose "economic

help to relatives in the past 12 months" as the IV of social network. Moreover, "post and tele-communications fee" was used to measure the social network, and was used instead of "gift of money" as the independent variable in the robustness analysis.

**3.3.3 Control variables.**   Control variables included the characteristics of the household head, family, and province. Referring to the research of Liu [66], the present study defined the "financial respondent" as the "household head" and, by default, the decision-maker and person in charge of major family affairs. The measured characteristics of the household head included gender, age, educational background, marriage and working status. The educational background of household head is assigned from 1 to 8, according to illiterate/semi-illiterate, elementary school, junior high school, senior high school/technical school/vocational high school, junior college, undergraduate, master's degree and doctor's degree. Family characteristics included family size, family net income, family assets, and urban-rural classification. The province characteristic included province location.

## 3.4 Model building

**3.4.1 Benchmark regression model.**   We used the OLS method to analyze the influence of social network on tourism consumption by Chinese families. The econometric regression model was designed as follows:

$$y_i = \beta lnsn_i + \gamma x_i + \varepsilon_i \tag{1}$$

where, $y$ represents the family tourism consumption expenditure, $lnsn$ is the explanatory variable that represents the social network, and $x$ is the control variables. $i$ represents the household. $\varepsilon_i$ is the error term.

**3.4.2 Quantile regression model.**   We used the QRM to analyze the changing characteristics of social network influence on different tourism consumption levels. The QRM was established as follows:

$$Q_q\left(\frac{y_i}{ln\,sn_i}, x_i\right) = \alpha_q + \beta_q\,ln\,s\,n_i + \gamma_q x_i + \varepsilon_i \tag{2}$$

where $Q_q\left(\frac{y_i}{ln\,sn_i}, x_i\right)$ represents that $y_i$ is the conditional quantile; $\alpha_q$, $\beta_q$ and $\gamma_q$ are the $q$ quantile regression coefficients, and $\varepsilon_i$ is the error term.

Descriptions of these variables are presented in Table 1.

# 4. Results and discussion

## 4.1 Benchmark results

Table 2 shows the regression results of the influence of the social network on households' tourism consumption. Column (1) in Table 2 is without the control variables. The social network coefficient was significantly positive at the 1% level, which revealed a positive association between social network and households' tourism consumption. To decrease the deviation of missing variables, there were three kinds of control variables, namely, characteristics of the head of the household, family and province. Control variables of characteristics of the head of the household are added to column (2). The relative control variables of family characteristics were added to column (3). All three kinds of control variables were added to column (4). For all chosen control variables, the social network coefficient was significant at the 1% level, which indicates that the social network significantly improved the tourism consumption level of Chinese families. The column (2) showed that for every 1% increase in the social network intensity, the tourism consumption expenditure of Chinese households increased by

**Table 1. Summary statistics.**

| Variable | Description | Obs. | Mean | S.D. | Min. | Max. |
|---|---|---|---|---|---|---|
| lnftc | Logarithm of household tourism consumption. | 3254 | 7.6596 | 1.3970 | 2.1972 | 11.9184 |
| lnsn | Logarithm of gift money expenditure. | 3254 | 8.2624 | 0.9781 | 3.9120 | 11.5129 |
| Gender | Gender of household head, male = 1, female = 0. | 3254 | 0.5080 | 0.5000 | 0 | 1 |
| Age | Age of household head. | 3254 | 45.5009 | 14.7284 | 18 | 90 |
| Education | Educational background of household head. | 3254 | 3.7074 | 1.4723 | 1 | 8 |
| Martial | Marriage of household head, married = 1, others = 0. | 3254 | 0.7938 | 0.4046 | 0 | 1 |
| Employment status | Working of household head, working = 1, others = 0. | 3254 | 0.7526 | 0.4316 | 0 | 1 |
| lnthni | Logarithm of family income. | 3254 | 11.4566 | 0.9189 | 0 | 15.9197 |
| lnfa | Logarithm of family assets. | 3254 | 8.8835 | 4.3027 | 0 | 15.4250 |
| Size | Family size. | 3254 | 3.4416 | 1.7540 | 1 | 15 |
| Urban | Urban and rural classification, urban = 1, rural = 0 | 3254 | 0.7277 | 0.4452 | 0 | 1 |
| Area | Province dummy variable, east and central = 1, west = 0. | 3254 | 0.7763 | 0.4168 | 0 | 1 |
| lnhelp | Logarithm of economic help to relatives | 3254 | 3.5418 | 4.1465 | 0 | 12.2060 |
| lnpt | Logarithm of post and telecommunications fee | 3254 | 7.7789 | 1.0456 | 0 | 11.0021 |

Source: Authors' estimation.

approximately 37%, which demonstrates the economic significance of the tourism consumption promotion effect of the social network. The regression coefficient of the social network in column (3) was 0.2718; therefore, the tourism consumption expenditure of families with a social network was 27.18% higher than that of families without a social network on average. In column (4) in Table 2, all control variables were added; the regression coefficient of the social network decreases, and the $R^2$ value increased compared with the results shown in column (2). However, for every 1% increased in the social network strength, the tourism consumption expenditure of Chinese households increased by approximately 28%, and thus the research conclusions were the same. Therefore, H1 was verified.

Other control variables also influenced family tourism consumption. A review of the household head's characteristics showed that gender negatively affected the tourism consumption expenditure of Chinese households at the 5% significance level, which indicates that female-headed households were more willing to spend money on tourism. Usually, women have more initiative in tourism decision making. Other than gender, age positively influenced family tourism consumption expenditure at the 1% significance level. For educational background, the educational level of the household head significantly promoted family tourism consumption. Compared to families with working heads, families with unemployed heads spent more on tourism, likely because most unemployed heads of households are retired, and retirement positively influenced tourism consumption expenditure [67]. The CFPS (2018) showed that 62% of families included in the present study had reached retirement age. From the perspective of family characteristics, the total income and assets positively influence family travel spending at the 1% significance level, which indicates that the higher the total household income, the greater the family assets, and the higher the family tourism consumption expenditure. Compared with rural families, urban families spent more on tourism, which might be because urban households had a higher economic ability and stronger willingness to travel than rural households. Moreover, the regression coefficient of family size was significantly negative at the 5% level. The larger the family size, the lower the household tourism consumption. One possible reason for this is that family population expansion leads to an increase in other living expenses, thus reducing the budget for family tourism consumption.

**Table 2. Impacts of social networks on household tourism consumption.**

| Variable | (1) | (2) | (3) | (4) |
|---|---|---|---|---|
| lnsn | 0.4157*** | 0.3742*** | 0.2718*** | 0.2750*** |
| | (0.0250) | (0.0246) | (0.0276) | (0.0275) |
| Gender | | -0.0727 | -0.0786* | -0.0885** |
| | | (0.0459) | (0.0438) | (0.0437) |
| Age | | 0.0111*** | 0.0102*** | 0.0104*** |
| | | (0.0020) | (0.0020) | (0.0020) |
| Education | | 0.3271*** | 0.1976*** | 0.2030*** |
| | | (0.0171) | (0.0189) | (0.0192) |
| Martial | | 0.0557 | -0.0314 | -0.0208 |
| | | (0.0600) | (0.0598) | (0.0599) |
| Employment status | | -0.1882*** | -0.1016 | -0.1059* |
| | | (0.0635) | (0.0612) | (0.0612) |
| lnthni | | | 0.3999*** | 0.4067*** |
| | | | (0.0528) | (0.0533) |
| lnfa | | | 0.0156*** | 0.0168*** |
| | | | (0.0054) | (0.0054) |
| Size | | | -0.0292** | -0.0340** |
| | | | (0.0139) | (0.0140) |
| Urban | | | 0.3632*** | 0.3744*** |
| | | | (0.0529) | (0.0528) |
| Area | | | | -0.1713*** |
| | | | | (0.0515) |
| Constant | 4.2252*** | 2.9828*** | -0.5137 | -0.5184 |
| | (0.2080) | (0.2262) | (0.4670) | (0.4720) |
| $R^2$ | 0.0847 | 0.1812 | 0.2641 | 0.2665 |
| N | 3254 | 3254 | 3254 | 3254 |

Source: Authors' estimation.

Note

***significance at the 1% level

** significance at the 5% level

* significance at the 10% level; standard errors are reported in parentheses; $R^2$: The coefficient of determination; N: The number of observations.

## 4.2 Endogenous treatment

There are some issues related to endogenous problems, and reverse causality cannot be ruled out. Social network might promote family tourism consumption, but people might also strengthen emotional connections or broaden their support circle by traveling. However, one's social network might be affected by some variables not included in the present study that are associated with family tourism consumption. To solve these problems, we used the IV method to re-estimate the model. After repeated tests, we chose "economic help to relatives in the past 12 months" as the IV of social network. From a correlation perspective, compared with families without financial assistance to relatives, families with financial assistance to relatives have a broader social scope. From an externality perspective, there is no direct relationship between a family's financial help to relatives and their travel decisions and consumption. Therefore, we considered that "economic assistance to relatives in the past 12 months" to be a suitable IV.

To verify the validity of the tool variables, we used the two-stage least squares method to perform regression analysis (see Table 3). The first stage regression results strongly rejected

**Table 3. Instrumental variable regression results.**

| Two-stage regression results | |
|---|---|
| lnsn | 0.5429* |
| | (0.2987) |
| Control variables | YES |
| Constant | -1.6583 |
| | (1.2916) |
| $R^2$ | 0.2351 |
| N | 3254 |
| **One-stage regression results** | |
| The coefficient of instrumental variable | 0.0181*** |
| P value of instrumental variable | 0.000 |
| F statistical value | 19.61 |
| Kleibergen-Paap rk LM statistic | 19.364 |
| LM statistical value | 0.0000 |
| Kleibergen-Paap rk Wald F statistic | 19.61 |

Source: Authors' estimation.

Note

\*\*\*significance at the 1% level

\*\* significance at the 5% level

\* significance at the 10% level; standard errors are reported in parentheses; $R^2$: The coefficient of determination; N: The number of observations.

the original hypothesis that IVs are unrecognizable, whereas the F-test results showed no weak IV problem. The second stage regression results showed that the coefficient of the social network was significantly positive at the 10% level. Therefore, even when correcting for endogenous problems, the social network still significantly increases the tourism consumption expenditure of Chinese households. Thus, H1 was further verified.

## 4.3 Robustness test

**4.3.1 Replacing explanatory variables.** In the present study, "post and telecommunications fee" was used to measure the social network, although the logarithmic value of "one year's post and telecommunications fee" was used instead of "gift of money" for the regression analysis. The regression results were shown in Table 4. Column (1) in Table 4 was without the control variables. The relative control variables of family characteristics of the head of the household are added to column (2). The relative control variables of family characteristics were added to column (3). All three kinds of control variables are added to column (4). Whatever model was adopted, the social network coefficient is significantly positive at the 1% level, which indicates that the post and telecommunications fee significantly increased the tourism consumption expenditure of Chinese households. Taking column (4) in Table 4 as an example, a 1% increase in household post and telecommunications costs caused a 36.67% increase in their tourism consumption expenditure. Therefore, the robustness test results also support H1.

**4.3.2 Smoothing singular values of samples.** Family income depends on resource endowment. The differences in resource endowments of different households mean that the income levels of families are different. Therefore, the microdata obtained by taking families or households as survey objects caused abnormal values in household income survey data due to differences in family resource endowments. However, households might overestimate or

**Table 4. Regression results for the replacement of explanatory variables.**

| Variable | (1) | (2) | (3) | (4) |
|---|---|---|---|---|
| lnpt | 0.2649*** | 0.2655*** | 0.1774** | 0.1756*** |
|  | (0.0332) | (0.0319) | (0.0402) | (0.0316) |
| Characteristics of household head |  | YES | YES | YES |
| Characteristics of family |  |  | YES | YES |
| Characteristics of province |  |  |  | YES |
| Constant | 5.5988*** | 3.6411*** | -0.1892 | -0.1624 |
|  | (0.2627) | (0.2842) | (0.5292) | (0.5318) |
| $R^2$ | 0.0393 | 0.1522 | 0.2461 | 0.2476 |
| N | 3254 | 3254 | 3254 | 3254 |

Source: Authors' estimation.

Note

***significance at the 1% level

** significance at the 5% level

* significance at the 10% level; standard errors are reported in parentheses; $R^2$: The coefficient of determination; N: The number of observations.

underestimate their overall household income due to personal or objective factors, which could result in the first singular value of the sample. The existence of outliers decreases the estimation efficiency of the OLS. Therefore, we applied lessons from the minorizing method and deleted the 5% outliers before and after the sample bases on the household income level to eliminate any adverse effects of outliers on the regression results. Column (1) in Table 5 is without the control variables. The relative control variables of characteristics of the head of the household have been added to column (2). The relative control variables of family characteristics have been added to column (3). All three kinds of control variables are added to column (4). As shown in Table 5, after smoothing the abnormal values of samples, although the coefficient of core explanatory variable, social network, decreased, it was still significantly positive at the 1% level. Therefore, the robustness test results support H1.

**Table 5. Regression results of singular values of smoothing samples.**

| Variable | (1) | (2) | (3) | (4) |
|---|---|---|---|---|
| lnsn | 0.3468*** | 0.3119*** | 0.2215*** | 0.2250*** |
|  | (0.0264) | (0.0263) | (0.0259) | (0.0259) |
| Characteristics of household head |  | YES | YES | YES |
| Characteristics of family |  |  | YES | YES |
| Characteristics of province |  |  |  | YES |
| Constant | 4.7890*** | 3.5395*** | -2.2880*** | -2.3328*** |
|  | (0.2194) | (0.2390) | (0.4500) | (0.4460) |
| $R^2$ | 0.06010 | 0.1532 | 0.2360 | 0.2408 |
| N | 2928 | 2928 | 2928 | 2928 |

Source: Authors' estimation.

Note

***significance at the 1% level

** significance at the 5% level

* significance at the 10% level; standard errors are reported in parentheses; $R^2$: The coefficient of determination; N: The number of observations.

**Table 6. Estimations of income class.**

| Variable | Low-income group | High-income group |
|---|---|---|
| lnsn | 0.2674*** | 0. 2725*** |
|  | (0.0330) | (0.0392) |
| Control variables | YES | YES |
| R$^2$ | 0.1566 | 0.1938 |
| N | 1797 | 1457 |

Source: Authors' estimation.

Note

***significance at the 1% level, ** significance at the 5% level, * significance at the 10% level; standard errors are reported in parentheses; R$^2$: The coefficient of determination. N: The number of observations.

## 4.4 Subsequent analysis

Considering the differences in income classes, family life cycle stages, urban and rural areas, and geographical locations, the tourism consumption expenditure of Chinese households can be expected to respond differently to social networks. To more accurately and carefully describe the influence of social networks on family tourism consumption, we applied an extensive analysis of the heterogeneous effects of social networks on the tourism consumption of Chinese households.

**4.4.1 Income class.** Table 6 shows the regression results of grouping Chinese households based on their income level. Based on the total net income of families, the samples are divided into low-income and high-income groups. The results show that the social network has different effects on tourism consumption in low-income and high-income families. Specifically, at the 1% significance level, the regression coefficient of the social network to tourism consumption of low-income families is 0.2674, whereas for high-income families it is 0.2725. Therefore, the social network has a stronger effect on the tourism consumption of the high-income class, and H2 is verified. The reason for this difference may be that income is the key factor determining households' tourism demand and consumption. Thus, the consumption ability of low-income families is restricted.

**4.4.2 Family life cycle.** Table 7 shows the regression results according to the different family life cycle stages. Based on the research of Wang et al. [68], we used the age of the household head to as a measure of the family life cycle. The sample families were divided into five

**Table 7. Estimated results of the household life cycle.**

| Variable | <25 | 25~34 | 35~44 | 45~54 | >54 |
|---|---|---|---|---|---|
| lnsn | 0.2493 ** | 0.2411*** | 0.4167*** | 0.2305*** | 0.2366*** |
|  | (0.1126) | (0.0432) | (0.0608) | (0.0537) | (0.0530) |
| Control variables | YES | YES | YES | YES | YES |
| R$^2$ | 0.2300 | 0.2716 | 0.3803 | 0.3000 | 0.2535 |
| N | 155 | 798 | 655 | 718 | 928 |

Source: Authors' estimation.

Note

***significance at the 1% level

** significance at the 5% level, * significance at the 10% level; standard errors are reported in parentheses; R$^2$: The coefficient of determination; N: The number of observations.

**Table 8. Estimated results of the urban–rural differentiation.**

| Variable | rural | urban |
|---|---|---|
| lnsn | 0.2248*** | 0.2989*** |
|  | (0.0521) | (0.0311) |
| Control variables | YES | YES |
| R$^2$ | 0.1529 | 0.2408 |
| N | 886 | 2368 |

Source: Authors' estimation.

Note

***significance at the 1% level, ** significance at the 5% level, * significance at the 10% level; standard errors are reported in parentheses; R$^2$: The coefficient of determination; N: The number of observations.

groups. The results showed that irrespective of age group, the regression coefficient of the social network was significantly positive at least at the 5% level. Among the five groups, the social network had the greatest influence on tourism consumption in household with a head aged 35–44 years, with a regression coefficient of 0.4166. This is consistent with Zhang's results [69]. Thus, the social network has a different effect on the tourism consumption of families of different age groups. Therefore, H3 was verified.

**4.4.3 Urban-rural differentiation.** Table 8 shows the regression results according to urban and rural classification. The social network had a significant effect on the tourism consumption expenditure of rural and urban families at the 1% significance level. The positive effect of the social network on family tourism consumption was different between urban and rural areas; therefore, H4 was supported. There are several possible explanations for this phenomenon. On the one hand, urban areas have excellent infrastructure, convenient transportation, and wide coverage of information networks, whereas these conditions are lacking in rural areas. Whether visiting relatives and friends directly or using mobile phones to contact them, urban areas are more convenient and conducive to expanding social networks. On the other hand, urban households still occupy the dominant position in China's tourism market as they have stronger economic position and willingness to travel; in contrast, rural households have lower income and less inclination to travel, which weakens the influence of the social network on the tourism consumption of rural households.

**4.4.4 Different regions.** Table 9 shows the regression results according to region. The regression coefficients of social networks for tourism consumption of Chinese households in

**Table 9. Estimated results by subregion.**

| Variable | Western region | Central region | Eastern region |
|---|---|---|---|
| lnsn | 0.2695*** | 0.2321*** | 0.3206*** |
|  | (0.0650) | (0.0476) | (0.0369) |
| Control variables | YES | YES | YES |
| R$^2$ | 0.2621 | 0.2346 | 0.2914 |
| N | 726 | 916 | 1612 |

Source: Authors' estimation.

Note

***significance at the 1% level, ** significance at the 5% level, * significance at the 10% level; standard errors are reported in parentheses; R$^2$: The coefficient of determination; N: The number of observations.

**Table 10. Results of quantile regression.**

| Variable | 10% | 30% | 50% | 70% | 90% |
|---|---|---|---|---|---|
| lnsn | 0.2936*** | 0.2841*** | 0.2617*** | 0.2525*** | 0.2047*** |
|  | (0.0404) | (0.0358) | (0.0332) | (0.0280) | (0.0467) |
| Control variables | YES | YES | YES | YES | YES |
| $R^2$ | 0.1293 | 0.1282 | 0.1564 | 0.1707 | 0.1659 |
| N | 3254 | 3254 | 3254 | 3254 | 3254 |

Source: Authors' estimation.

Note

***significance at the 1% level, ** significance at the 5% level, * significance at the 10% level; standard errors are reported in parentheses; $R^2$: The coefficient of determination; N: The number of observations.

the eastern, central, and western regions were significantly positive at the 1% level. Thus, the social network had a positive effect on the tourism consumption of households in the eastern, central and western regions; however, there are differences. The eastern region was more strongly affected than the central and western regions. Therefore, H5 was supported. This may be because that there is a higher level of economic development in the eastern region than that in the central and western regions, and the eastern region thus has more advantages in terms of politics, economy, science and technology, culture, and education. Therefore, families in the east may have more confidence and convenience to travel than those in the other regions.

**4.4.5 Quantile regression model.** As shown in Table 10, the regression coefficients of social networks at each locus were significantly positive at the 1% level. Thus, social networks significantly increase household tourism consumption expenditure, whether at low, middle, or high points. However, with the increase of quantile, the quantile regression coefficient of the social network showed a decreasing trend. Therefore, the positive effect of the social network in families with a lower tourism consumption level was greater than that in families with a higher tourism consumption level. Specifically, at 10%, 30%, and 50% loci, the regression coefficients of the social network to family tourism consumption decreased continuously to 0.2902, 0.2859, and 0.2615, respectively, and then continued to decrease, and the marginal effect decreased to 0.2075 at the 90% loci. Although the social network increased the tourism consumption of households at each locus by 20.8%–29.0%, the influence of the social network on the tourism consumption of Chinese households gradually decreased with continuous improvement in the level of household tourism consumption. Therefore, H6 was verified.

# 5. Conclusions and policy implications

## 5.1 Conclusions

Based on the perspective of social economics, employing OLS, IV and QRM, this paper uses the data of 3,254 samples from the CFPS (2018) to examine the effect of social network on tourism consumption by Chinese households. All six research hypotheses were supported. The main conclusions will now be discussed.

The OLS results showed that the social network promotes Chinese households' tourism consumption, which increased by about 28% for every 1% increase in social network strength. It is further confirmed using IV to address the issue of endogenous social network formation, as well as by other robustness checks.

In addition, subsequent analyses showed that the impact of social networks on tourism consumption by Chinese households was heterogeneous. The positive effect of social network on family tourism consumption was significant in both low-income and high-income families,

but the impact was stronger in high-income families. From the perspective of family life cycle, the social network had a significant impact on tourism consumption of families of different age groups, although there were differences between age groups. Moreover, the social network promoted family tourism consumption in both urban and rural areas, and had a stronger influence on tourism consumption of urban households. Also, the positive impact of social network on family tourism consumption in the eastern region was higher than that in the central and western regions.

Finally, the results of QRM showed that the influence of social network on tourism consumption by Chinese households gradually decreased with the improvement of the tourism consumption level, which indicates that the positive effect of social network was more evident in groups with lower tourism consumption.

In short, the existence of a high correlation between social network and tourism expenditure by Chinese households, which means that social network should be considered as a vital resource for promoting household tourism consumption, and enhancing residents' sense of acquisition and happiness.

## 5.2 Policy implications

These research results have some policy implications.

First, the social network had a significant positive impact on tourism consumption by Chinese households, which was generated through the demonstration effect and the linkage effect. Therefore, the social network is an important avenue by which to increase family tourism consumption. In light of this, the government should consider strengthening the establishment of different forms of social platforms, building a good social network relationship, and stimulating residents' desire for tourism consumption. If we give full play to the role of grassroots organizations, this can guide residents to form modern consumption concepts, cultivate a rational tourism consumption decision-making ability, and promote the rationalization of residents' tourism consumption through the consumption linkage and demonstration effects. Moreover, it could help households to engage in green and low-carbon tourism consumption and cultivate good family tourism consumption habits.

Second, the social network also affected Chinese family tourism consumption through the income effect and the security effect, and there were differences in this effect between different income groups. Therefore, the government should strengthen social security for low-income groups to ensure their basic living needs and improve their ability to pay for tourism consumption. At the same time, tourism consumption capacity is relatively strong in high-income groups, and they pay more attention to the construction of contacts in their social networks. Therefore, the government should formulate reasonable tourism consumption policies to encourage high-income groups to avoid excessive conspicuous tourism consumption and the subsequent formation of a bad social atmosphere. In addition, the "leader" role of high-income groups in consumption could be used to drive other consumer groups to engage in reasonable tourism consumption.

Finally, although the social network significantly increased the level of residents' consumption, the negative news about tourism consumption in residents' social network is also worthy of attention. For products and services that harm the rights and interests of tourists, the social network will have a negative impact on tourist destinations or household consumption of such products and services. Therefore, the government should strengthen the supervision of tourism service quality and the protection of tourists' rights and interests, improve the quality of tourism service products, and minimize the impact of the social network on negative tourism events.

### 5.3 Scope for future research

Although the results of this study are of immense importance to policymakers, this research inevitably bears several limitations, that should be addressed in future research.

Firstly, the social network has complex and "privacy" characteristics, which makes the measurement of social network indicators in this study slightly insufficient in terms of representativeness and accuracy. Therefore, future work should take more accurate measurements of social networks.

Secondly, there are many subdivisions of tourism consumption expenditure, such as domestic tourism consumption and foreign tourism consumption, and tourism consumption expenditure can also be subdivided into the expenses for tourism transportation, accommodation, catering, and health, etc. In the future, researchers should conduct more nuanced studies that focus on the relationship between the social network and the segments of tourism consumption by Chinese households.

Moreover, the expenditure of household tourism consumption is affected by many other macroeconomic and social factors, such as consumption environment, macroeconomic situation, and air quality condition. Due to data limitations, these factors were not considered in this study, which may have affected the research conclusions to a certain extent. The present study could be expanded by incorporating these influencing factors into the research framework.

## Supporting information

**S1 Data.**
(XLSX)

## Acknowledgments

The authors would like to thank the Institute of Social Science Survey (ISSS) of Peking University for providing data of CFPS. The authors wish to thank the anonymous reviewers whose comments helped to improve the paper.

## Author Contributions

**Conceptualization:** Genjin Sun, Dan Wang.

**Data curation:** Yanxiu Liu.

**Methodology:** Genjin Sun, Dan Wang.

**Writing – original draft:** Genjin Sun, Qi Qian, Dan Wang.

**Writing – review & editing:** Genjin Sun, Bo Pu, Dan Wang.

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
