## [Decision Letter · Decision Letter 0]

11 Jul 2022

PONE-D-22-15535Social Network and Tourism Consumption by Households: An Evidence from ChinaPLOS ONE

Dear Dr. sun,

Thank you for submitting your manuscript to PLOS ONE. After careful consideration, we feel that it has merit but does not fully meet PLOS ONE’s publication criteria as it currently stands. Therefore, we invite you to submit a revised version of the manuscript that addresses the points raised during the review process.

We look forward to receiving your revised manuscript.

Kind regards,

Jun Yang

Academic Editor

PLOS ONE

Journal Requirements:

The author(s) received no specific funding for this work.Include this sentence at the end of your statement: The funders had no role in study design, data collection and analysis, decision to publish, or preparation of the manuscript. 

Additional Editor Comments:

Major Revision

Reviewers' comments:

Reviewer's Responses to Questions

**Comments to the Author**

1. Is the manuscript technically sound, and do the data support the conclusions?

Reviewer #1: Yes

Reviewer #2: Yes

2. Has the statistical analysis been performed appropriately and rigorously? 

Reviewer #1: Yes

Reviewer #2: Yes

3. Have the authors made all data underlying the findings in their manuscript fully available?

Reviewer #1: Yes

Reviewer #2: Yes

4. Is the manuscript presented in an intelligible fashion and written in standard English?

Reviewer #1: Yes

Reviewer #2: No

5. Review Comments to the Author

Reviewer #1: 1. Please highlight the key points in the abstract.

2.The introduction should further highlight what is the motivation of the paper.

3. The literature review section is also not focused enough. Please read Effects of rural revitalization on rural tourism. Journal of Hospitality and Tourism Management (2021),https://doi.org/10.1016/j.jhtm.2021.02.008.

An input-output analysis of transportation equipment manufacturing industrial transfer: Evidence from Beijing-Tianjin-Hebei region, China.Growth and Change (2021).https://doi.org/10.1111/grow.12571

The influence of high-speed rail on ice–snow tourism in northeastern China. Tourism Management（2020）, doi:10.1016/j.tourman.2019.104070.

4. It is suggested to update the data of the paper, so that the results may be more meaningful.

5. Drawing lessons from the study by Yi et al. How did you draw this lessons ?

6. The results should be better described, discussed and justified.

Reviewer #2: The manuscript topic is relevant and valuable to existing research. It is readable and could be well referenced by chinese readers. Here are some commments for the authors from my perspective. 1、the language of the paper should be further improved. 2、why does the authors show Quantile Regression Model in the part 4.4.5? I think it should be moved to the methology part. 3、There is no discussion in the paper, which weaken the theoretical contribution of the research. 4、I think implication is a better word than recommendations.

6. PLOS authors have the option to publish the peer review history of their article (what does this mean?). If published, this will include your full peer review and any attached files.

Reviewer #1: No

Reviewer #2: No

---

## [Author Response · Author response to Decision Letter 0]

12 Aug 2022

Dear Sir:

Now we submit Manuscript, Revised Manuscript with Track Changes, Response to Reviewers as a separate file, respectively.

During the modification process, we followed journal requirements and reviewer comments. We ensured that our manuscript meets PLOS ONE's style requirements. 

The authors received no specific funding for this work. The funders had no role in study design, data collection and analysis, decision to publish, or preparation of the manuscript.

We added our full ethics statement in the ‘Methods’ section of manuscript file, in which we indicated the IRB (00001052-14010) and the Biomedical Ethics Committee of Peking University who approved our study, as well as we obtained informed written consent. We ensured references were complete and correct.

Meanwhile, we entrust the Bioedit Ltd Company for editing proper English language, grammar, punctuation, spelling, and overall style of this manuscript. The editing certificate is available to view at the following URL:

[https://www.bioedit.com/digitalcertificate/view/e6cfff2213131023620e1705a516a6ecf1341300]

---

## [Decision Letter · Decision Letter 1]

19 Sep 2022

Social network and tourism consumption by households: Evidence from China

PONE-D-22-15535R1

Dear Dr. sun,

We’re pleased to inform you that your manuscript has been judged scientifically suitable for publication and will be formally accepted for publication once it meets all outstanding technical requirements.

Kind regards,

Jun Yang

Academic Editor

PLOS ONE

Additional Editor Comments (optional):

Accept

Reviewers' comments:

Reviewer's Responses to Questions

**Comments to the Author**

1. If the authors have adequately addressed your comments raised in a previous round of review and you feel that this manuscript is now acceptable for publication, you may indicate that here to bypass the “Comments to the Author” section, enter your conflict of interest statement in the “Confidential to Editor” section, and submit your "Accept" recommendation.

Reviewer #1: (No Response)

Reviewer #2: All comments have been addressed

2. Is the manuscript technically sound, and do the data support the conclusions?

Reviewer #1: (No Response)

Reviewer #2: Yes

3. Has the statistical analysis been performed appropriately and rigorously? 

Reviewer #1: (No Response)

Reviewer #2: Yes

4. Have the authors made all data underlying the findings in their manuscript fully available?

Reviewer #1: (No Response)

Reviewer #2: Yes

5. Is the manuscript presented in an intelligible fashion and written in standard English?

Reviewer #1: (No Response)

Reviewer #2: Yes

6. Review Comments to the Author

Reviewer #1: The authors have adequately addressed comments raised in a previous round of review and I feel that this manuscript is now acceptable for publication.

Reviewer #2: I am glad to review the manuscript, i think the manuscript is well written and the topic is interesting to potentical readers. I suggested accept after minor revision. The following comments can be refferened to improve the manuscript.

1. The language should be further improved by native speakers.

2. The introduction part should introduce the research gap more clearly.

3.Theoretical contributions should mentioned in the disscussion part.

7. PLOS authors have the option to publish the peer review history of their article (what does this mean?). If published, this will include your full peer review and any attached files.

Reviewer #1: No

Reviewer #2: No

---

## [Editor Report · Acceptance letter]

22 Sep 2022

PONE-D-22-15535R1 

Social network and tourism consumption by households: Evidence from China 

Dear Dr. Sun:

I'm pleased to inform you that your manuscript has been deemed suitable for publication in PLOS ONE. Congratulations! Your manuscript is now with our production department. 

Kind regards, 

on behalf of

Dr. Jun Yang 

Academic Editor

PLOS ONE